# Augmented Expression of the IL3RA/CD123 Gene in MLL/KMT2A-Rearranged Pediatric AML and Infant ALL

**Sanjive Qazi and Fatih M. Uckun ***

Immuno-Oncology Program, Ares Pharmaceuticals, St. Paul, MN 55110, USA
* Correspondence: fatih.uckun@aresmit.com

**Abstract:** Here, we evaluated transcript-level IL3RA/CD123 expression in mixed lineage leukemia 1 (MLL) gene/KMT2A-rearranged (MLL-R$^+$) vs. MLL-R$^-$ pediatric AML as well as infant ALL by comparing the archived datasets of the transcriptomes of primary leukemic cells from the corresponding patient populations. Our studies provide unprecedented evidence that IL3RA/CD123 expression exhibits transcript-level amplification in MLL-R$^+$ pediatric AML and infant ALL cells. IL3RA was differentially upregulated in MLL-AF10$^+$ (2.41-fold higher, $p$-value = 4.4 × 10$^{-6}$) and MLL-AF6$^+$ (1.83-fold higher, $p$-value = 9.9 × 10$^{-4}$) but not in MLL-AF9$^+$ cases compared to other pediatric AML cases. We also show that IL3RA/CD123 expression is differentially amplified in MLL-AF4$^+$ (1.76-fold higher, $p$-value = 2.1 × 10$^{-4}$) as well as MLL-ENL$^+$ infant ALL (1.43-fold higher, $p$-value = 0.055). The upregulated expression of IL3RA/CD123 in MLL-R$^+$ pediatric AML and infant ALL suggests that CD123 may be a suitable target for biotherapy in these high-risk leukemias.

**Keywords:** AML; ALL; IL3RA; CD123; MLL gene; KMT2A; bispecific antibody; CAR T-cells; leukemia

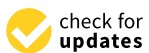

## 1. Introduction

The 90 kb lysine [K]-methyltransferase 2A (KMT2A)/mixed lineage leukemia 1 (MLL) gene located at 11q23 on the long arm of chromosome 11 contains 36 exons and encodes a 431 kDa protein involved in the regulation of transcription [1]. De novo acute myeloblastic leukemias (AML) and acute lymphoblastic leukemias (ALL) with rearrangements (r) of the MLL/KMT2A gene occur in infants, children, and adults [2–15]. MLL-R$^+$ AML and ALL are most frequent in infants. Over 100 MLL rearrangements involving >90 partner genes, including AFF1/AF4, MLLT3 (formerly AF9), and MLLT1/ENL, have been identified that generate leukemogenic fusion genes involving the N-terminus of *MLL/KMT2A* with the C-terminus of the partner gene that alter the transcriptome and epigenetic regulatory landscape and are frequently associated with a poor prognosis [16–18]. The most common fusion partners in AML are located on chromosomes 6q27 (MLLT4, or AF6), 9p22 (MLLT3, or AF9), 19p13.3 (MLLT1, or ENL), 19p13.1 (ELL), 19p13.3 (SH3GL1, or EEN), 16p13.3 (CREBBP, or CBP), and 22q13.2 (EP300, or p300) [17,18]. MLL rearrangements are also found in ALL, and the most common translocation partner in these cases is the AF4/FMR2 family, member 1 (AFF1) gene on chromosome 4q21 [17,18]. MLL rearrangements also occur in secondary or therapy-related acute leukemias [15].

Despite recent advances in risk-adjusted multi-modality therapy, precision medicine, biotherapy, and hematopoietic stem cell transplantation, AML and ALL patients with MLL-R$^+$ leukemia have disappointing treatment outcomes [1–22]. Therefore, developing effective standard treatments for MLL-R$^+$ acute leukemias remains an unmet and urgent medical need.

The α-chain of IL-3 receptor/CD123 has emerged as a biotherapeutic target in hematologic malignancies [23–26]. CD123 expression has been associated with an adverse prognosis in AML [27–30]. Several biotherapeutic agents targeting CD123 have been developed, such as the human IL3 fusion toxin Tagraxofusb, the antibody–drug conjugate

IMGN632, the bispecific CD3-engaging antibody APVO436, the bispecific dual-affinity re-targeting (DART) antibody Flotetuzumab, bispecific T-cell engagers (BiTEs) AMG673 and AMG330, bispecific and trispecific killer cell engagers, radioimmunoconjugates, and CD123-targeting CAR T-cells. Some of these agents have exhibited promising anti-leukemic activity in early clinical trials that tested their clinical potential in relapsed/refractory adult AML patients [24,25,31–36]. The expression of CD123 in MLL-R+ acute leukemias or the activity of CD123-directed biotherapeutic agents against MLL-R+ AML or ALL cells has not yet been studied.

The purpose of the present study was to evaluate CD123 expression in MLL-R+ pediatric AML and infant ALL. Our studies provide unprecedented evidence that IL3RA/CD123 expression exhibits transcript-level amplification in MLL-R+ pediatric AML and infant ALL cells. Taken together with the observed clinical activity of CD123-targeting biotherapeutic agents in adult AML patients, the presented data provide a scientific rationale for therapeutic targeting of CD123 in MLL-R+ pediatric AML and infant ALL patients.

## 2. Materials and Methods

### 2.1. Statistical Methods for Gene Chip Normalization for AML Samples

We performed a probeset-level normalization procedure to enable comparisons of gene expression levels of specific genes in leukemic blast cells from AML patients versus normal bone marrow hematopoietic cells, as previously described [37]. Raw Affymetrix .CEL data files were obtained from 3 datasets deposited in the NCBI repository (GSE13159, GSE19577, and GSE17855). These raw files were pre-processed for batch normalization by utilizing Aroma Affymetrix statistical packages (aroma.affymetrix_3.2.0, aroma.core_3.2.2, aroma.light_3.24.0, and affxparser_1.66.0) run in the R-studio environment (RStudio 2021.09. running with R version 4.1.2 R Foundation for Statistical Computing, Vienna, Austria, (1 November 2021)). The PM signals were quantified using robust multiarray analysis (RMA) in a 3-step process, including RMA background correction, quantile normalization, and summarization by a log-additive model of probes in a probeset across these samples (RmaPlm method adapted in Aroma Affymetrix). All expression values were $\log_2$-scaled for visualization in cluster figures. Sample annotations and patient group assignments were obtained from data files in the GEO repository: GSE17855_series_matrix.txt.gz and GSE19577_series_matrix.txt.gz, and then accessed using the programming utility, GEOquery_2.62.1 and stringr_1.4.0, implemented in the R environment. Gene symbol annotation for each probeset was obtained from the database provided by the Bioconductor repository for R software (http://www.bioconductor.org/ accessed on 8 July 2022) (hgu133plus2.db).

The normalized gene expression dataset used in our analyses included data on 279 pediatric patients with AML (GSE19577 and GSE17855). There were 89 pediatric patients with MLL-R+ AML. The control group for comparison in the "other" category (N = 190; GSE17855) for pediatric cases with MLL-R+ AML included the following subsets: cytogenetically normal (CN)-AML (N = 39); inv(16) (N = 27); t(15;17) (N = 19); t(7;12) (N = 7); t(8;21) (N = 28); remaining cytogenetics (N = 45); and unknown cytogenetics (N = 25). GSE13159 included 74 normal/non-leukemic bone marrow samples that were also used as a reference set in our comparisons.

### 2.2. Statistical Methods for Gene Chip Normalization for ALL Samples

We downloaded raw CEL files from 8 datasets deposited in the NCBI repository (GSE11877, GSE13159, GSE13351, GSE18497, GSE28460, GSE7440, GSE68720, and GSE32962, https://www.ncbi.nlm.nih.gov/geo/, accessed on 11 July 2022) to create a working database, and by performing a probeset-level normalization procedure, we were able to compare gene expression levels of specific genes in leukemic blast cells versus normal bone marrow hematopoietic cells from non-leukemic healthy volunteers.

The normalized gene expression dataset from the working database was interrogated to identify MLL-R+ samples, other MLL-R− samples, and normal/non-leukemic samples to compile a dataset for use in the analyses in this report. This dataset included data

on 201 adult patients with B-lineage ALL (GSE13159) and 97 infants with B-lineage ALL (GSE68720). There were 80 infants with MLL-R$^+$ ALL (partner genes: AF4 (N = 48); ENL (N = 16); AF9 (N = 6); ASAH3 (N = 1); EPS15 (N = 3); and unknown MLL partners (N = 6)). The infant MLL-R$^+$ ALL group included 2 subgroups of particular interest with specified partner genes that were each evaluated separately as well for differential expression of CD123: AF4 (N = 48) and ENL (N = 16). The control group for comparison in the "other" category for infants with MLL-R$^+$ ALL included 17 infants with MLL-R$^-$ (MLL germline/WT) ALL. GSE13159 included 74 normal/non-leukemic bone marrow samples that were also used as a reference set in our comparisons.

The probeset-level normalization algorithm [37] utilized perfect match (PM) signal values for probesets extracted from raw CEL files. Probe identifiers were matched to signal values using the Affymetrix CDF file (HG-U133_Plus_2, monocell.CDF) required to run the Aroma Affymetrix statistical package (version 0.97.551, R-studio Inc. (Boston, MA, USA), running with R 3.01). This probeset-level robust multiarray analysis (RMA) normalization procedure was performed across all samples by implementing 3 sets of calculations: RMA background correction, quantile normalization, and summarization by a log-additive model of probes in a probeset across these samples (RmaPlm method adapted in Aroma Affymetrix). All expression values were log$_2$-scaled for statistical comparisons and visualization in cluster figures. Two statistical parameters (RLE (Relative Log Expression) and NUSE (Normalized Unscaled Standard Error)) were utilized to assess the gene chip normalization procedure: for each probeset and each array, ratios were calculated between the expression of a probeset and the median expression of this probeset across all arrays of the database. Box plots of the RLE and NUSE were assessed to ensure they were within the quality bounds for all gene chips in the database. Gene symbol annotation for each probeset was obtained from the database provided by the Bioconductor repository for R software (http://www.bioconductor.org/, accessed on 8 July 2022) (hgu133plus2.db).

### 2.3. Statistical Methods for Differential Gene Expression

Our analyses of AML primarily focused on the expression levels of the CD13, CD33, and CD34 genes, used as AML-associated markers, and IL3RA/CD123. We also performed focused gene expression analyses comparing primary leukemic blasts from MLL-AF9$^+$ AML versus either MLL-AF10$^+$ AML or MLL-AF6$^+$ AML patients, interrogating the following control genes: CD13, CD14, CD33, and CD34, in addition to the following receptors for cytokines: CSF3R, KIT, CSF1R, IL2RA, IL6R, FLT3, IL1R1, CSF2RB, IL5RA, CSF2RA, and IL3RA. Our analyses of ALL focused on the expression levels of CD19 and CD34 genes, used as markers, and IL3RA/CD123. Mixed Model ANOVAs implemented in R version 4.1.2 (1 November 2021) were run in the RStudio 2021.09.0 Build 351 environment, as previously described [37,38]. We investigated differential gene expression levels in leukemia cells from MLL-R$^+$ subsets of high-risk acute leukemias vs. leukemia cells from MLL-R$^-$ subsets. The gene expression levels in leukemia cells from a specified subset were compared to the gene expression levels in normal hematopoietic cells, as well as the gene expression levels in leukemia cells from other subsets of leukemia patients, and the corresponding fold differences in expression were determined. The statistical model (built using lmerTest_3.1-3 and lme4_1.1-27.1 statistical packages implemented in Rstudio) controlled for variation caused by 2 fixed factors: "probeset" and "diagnostic group". The variation caused by the interaction term, "probeset $\times$ diagnostic group", was used to calculate least square means and standard error values to compare high-risk subsets of acute leukemia to the reference samples (normal or other). Gene-chip-to-gene-chip variation was accounted for by the random factor optimized utilizing the REML criterion (fits a variance-component model by residual (or restricted) maximum likelihood). The error term calculated for the interaction term was utilized to calculate statistical significance for comparison groups for each probeset using emmeans_1.7.0 and lsmeans_2.30-0 statistical packages. Differences with *p*-values less than 0.05 were deemed significant.

### 2.4. Hierarchical Clustering Analysis

We performed a two-way hierarchical clustering technique to organize expression patterns (mean-centered to $\log_2$-transformed RMA values of control samples) such that samples and probesets displaying similar expression profiles were grouped together using the average distance metric (default Euclidean distance implemented using the heatmap.2 function in the R package gplots_3.1.1), as previously described in detail [39,40].

## 3. Results

### 3.1. Augmented Expression of IL3RA/CD123 in Primary Leukemic Blasts from MLL-R+ Pediatric AML Patients

We first examined the transcript-level expression of IL3RA/CD123 in pediatric MLL-R+ AML cases. Primary leukemic blasts from pediatric patients with MLL-R+ AML (N = 89) expressed 1.88-fold higher levels of IL3RA/CD123 than normal hematopoietic cells in normal/non-leukemic control bone marrow samples (N = 74) ($p < 10^{-8}$) (Figure 1A). Notably, IL3RA/CD123 was expressed at 1.39-fold higher levels in primary leukemic blasts from the MLL-R+ pediatric AML subset (N = 89) vs. other subsets ($p = 3.3 \times 10^{-5}$) (N = 190) (Figure 1B).

We next sought to evaluate the potential impact of the specific partner genes AF10 (N = 10), AF6 (N = 11), and AF9 (N = 11) on IL3RA/CD123 upregulation in pediatric AML cases (Figure 2A–C). Primary leukemia cells from patients with MLL-AF10+ (3.3-fold higher, $p$-value $< 10^{-8}$), MLL-AF6+ (2.5-fold higher, $p$-value $= 10^{-7}$) and MLL-AF9+ (1.74-fold higher, $p$-value = 0.0011) fusions expressed significantly higher levels of IL3RA/CD123 than normal hematopoietic cells (Figure 2A–C left panels). IL3RA/CD123 was differentially upregulated in MLL-AF10+ (2.41-fold higher, $p$-value $= 4.4 \times 10^{-6}$) and MLL-AF6+ (1.83-fold higher, $p$-value $= 9.9 \times 10^{-4}$) but not in MLL-AF9+ pediatric cases compared to MLL-R− pediatric AML cases (N = 190) (Figure 2A–C right panels).

MLL-AF9+ AML cells had a 6.81-fold higher level of CD14 ($p < 1 \times 10^{-8}$), a 1.89-fold lower level of CD33 ($p$ = 0.016), and a 2.44-fold lower level of CD34 ($p$ = 0.0086) than MLL-AF10+ AML cells, consistent with a more mature myelo-monocytic/monocytic differentiation stage. Compared to MLL-AF10+ AML cells, their IL3RA/CD123 expression was 1.92-fold lower ($p$ = 0.014) (Figure 3A). MLL-AF9+ AML cells also had lower levels of CD34 (1.69-fold; $p$ = 0.058) and IL3RA/CD123 (1.45-fold; $p$ = 0.17) than MLL-AF6+ AML cells, but the differences were not statistically significant (Figure 3B).

We next sought to determine whether IL3RA/CD123 was the only cytokine receptor that was expressed at a lower level in MLL-AF9+ AML cells compared to MLL-AF10+ AML cells. As shown in Figure 4, receptors for FLT3 ligand (viz.: FLT3), GM-CSF (viz.: CSF2RA), and IL6 also showed transcript-level downregulation in MLL-AF9+ AML cells compared to MLL-AF10+ AML cells, whereas CSF2RB, the gene for the common beta chain of the high-affinity receptor for IL-3, IL-5, and CSF, was expressed at a 2.02-fold higher level (Figure 4). These results illustrate that the composite cytokine receptor profile of MLL-AF9+ AML cells differs from that of MLL-AF10+ AML cells across multiple cytokine receptors. Likewise, MLL-AF9+ AML cells expressed lower levels of not only IL3RA/CD123 but also FLT3, CSF2RA/GMCSF-RA, IL6R, CSF1R, and KIT compared to MLL-AF6+ AML cells (Figure 5).

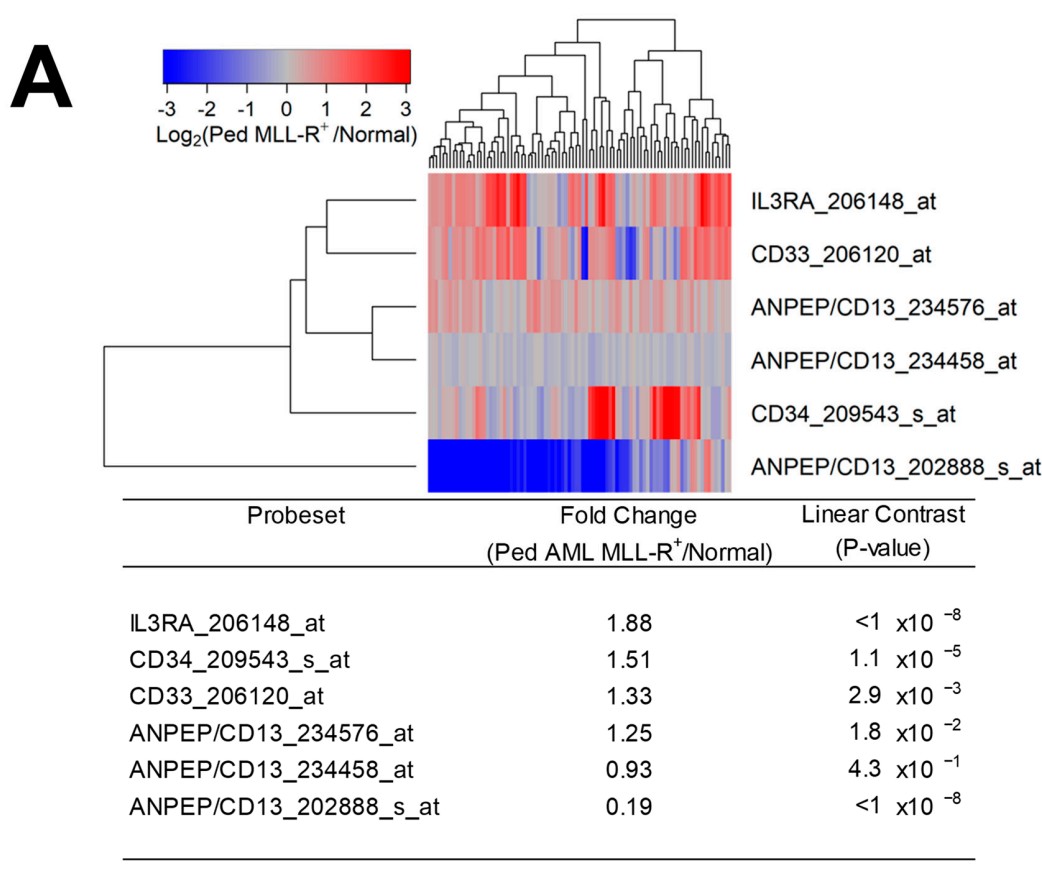

| Probeset | Fold Change (Ped AML MLL-R+/Normal) | Linear Contrast (P-value) |
|---|---|---|
| IL3RA_206148_at | 1.88 | $<1 \times 10^{-8}$ |
| CD34_209543_s_at | 1.51 | $1.1 \times 10^{-5}$ |
| CD33_206120_at | 1.33 | $2.9 \times 10^{-3}$ |
| ANPEP/CD13_234576_at | 1.25 | $1.8 \times 10^{-2}$ |
| ANPEP/CD13_234458_at | 0.93 | $4.3 \times 10^{-1}$ |
| ANPEP/CD13_202888_s_at | 0.19 | $<1 \times 10^{-8}$ |

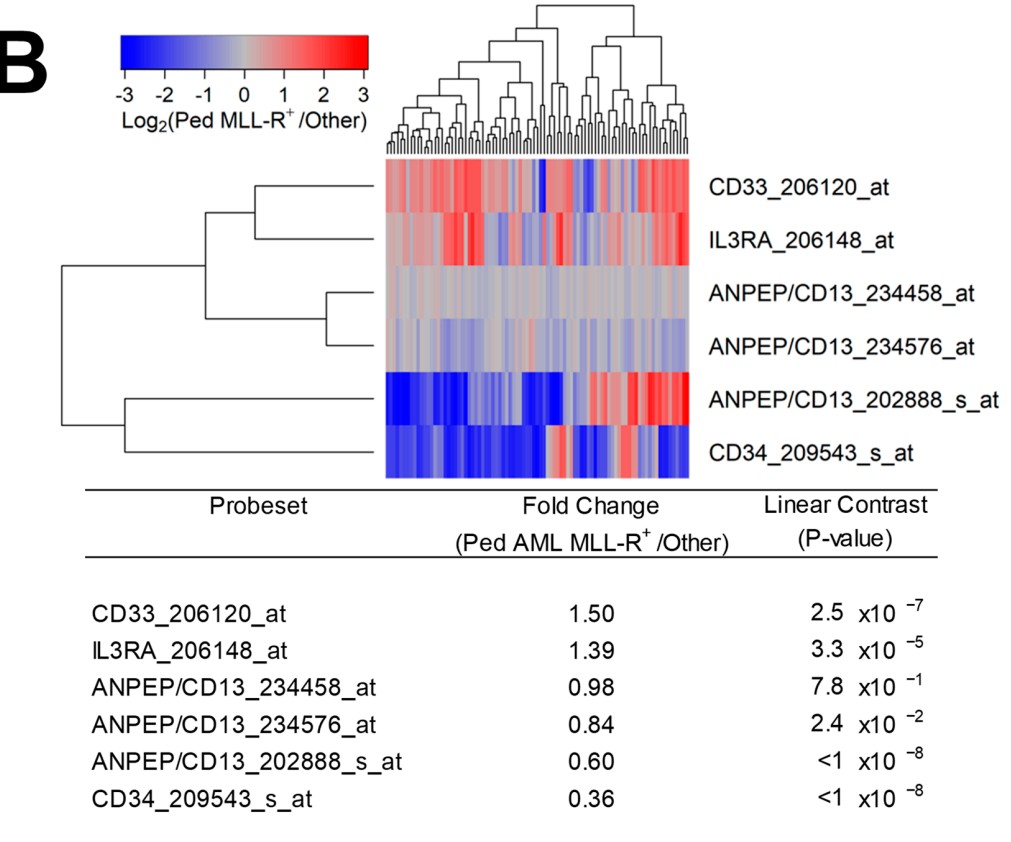

| Probeset | Fold Change (Ped AML MLL-R+/Other) | Linear Contrast (P-value) |
|---|---|---|
| CD33_206120_at | 1.50 | $2.5 \times 10^{-7}$ |
| IL3RA_206148_at | 1.39 | $3.3 \times 10^{-5}$ |
| ANPEP/CD13_234458_at | 0.98 | $7.8 \times 10^{-1}$ |
| ANPEP/CD13_234576_at | 0.84 | $2.4 \times 10^{-2}$ |
| ANPEP/CD13_202888_s_at | 0.60 | $<1 \times 10^{-8}$ |
| CD34_209543_s_at | 0.36 | $<1 \times 10^{-8}$ |

**Figure 1.** Amplified expression of IL3RA/CD123 in primary leukemic blasts from pediatric patients with MLL-R+ AML. We examined the gene expression data in the archived datasets GSE13159, GSE17855,

and GSE19577. The cluster figures display the expression levels in MLL-R$^+$ AML cells mean-centered to the reference group (normal bone marrow samples or other AML subsets) for log$_2$-transformed fold change values (blue represents underexpression and red represents overexpression in MLL-R$^+$ samples). The expression levels of coregulated probesets for both probesets (rows) and patients (columns) are organized in the depicted dendrograms. (**A**) Depicted are the differential gene expression changes in log$_2$-transformed robust multiarray analysis (RMA)-normalized values for 89 pediatric patients with MLL-R$^+$ AML (GSE17855 (N = 47) and GSE19577 (N = 42)). The expression levels in MLL-R$^+$ AML cells were mean-centered to the mean expression of 74 normal/non-leukemic control samples (GSE13159) and visualized using a two-way clustering algorithm. This analysis showed that IL3RA_206148_at was the most significantly upregulated probeset (fold change = 1.88; $p$-value < 10$^{-8}$) followed by CD34_209543_s_at (fold change = 1.51; $p$-value = 1.1 × 10$^{-5}$) and CD33_206120_at (fold change = 1.33; $p$-value = 0.0029). IL3RA/CD123 expression exhibited coregulation with CD33. (**B**) Comparing 190 cases of MLL-R$^-$ pediatric AML (GSE17855) with 89 cases of MLL-R$^+$ pediatric AML samples (GSE17855 (N = 47) and GSE19577 (N = 42)) showed that IL3RA_206148_at exhibited a 1.39-fold increase in expression ($p$-value = 3.3 × 10$^{-5}$). IL3RA and CD33 were coregulated as visualized in the cluster figure.

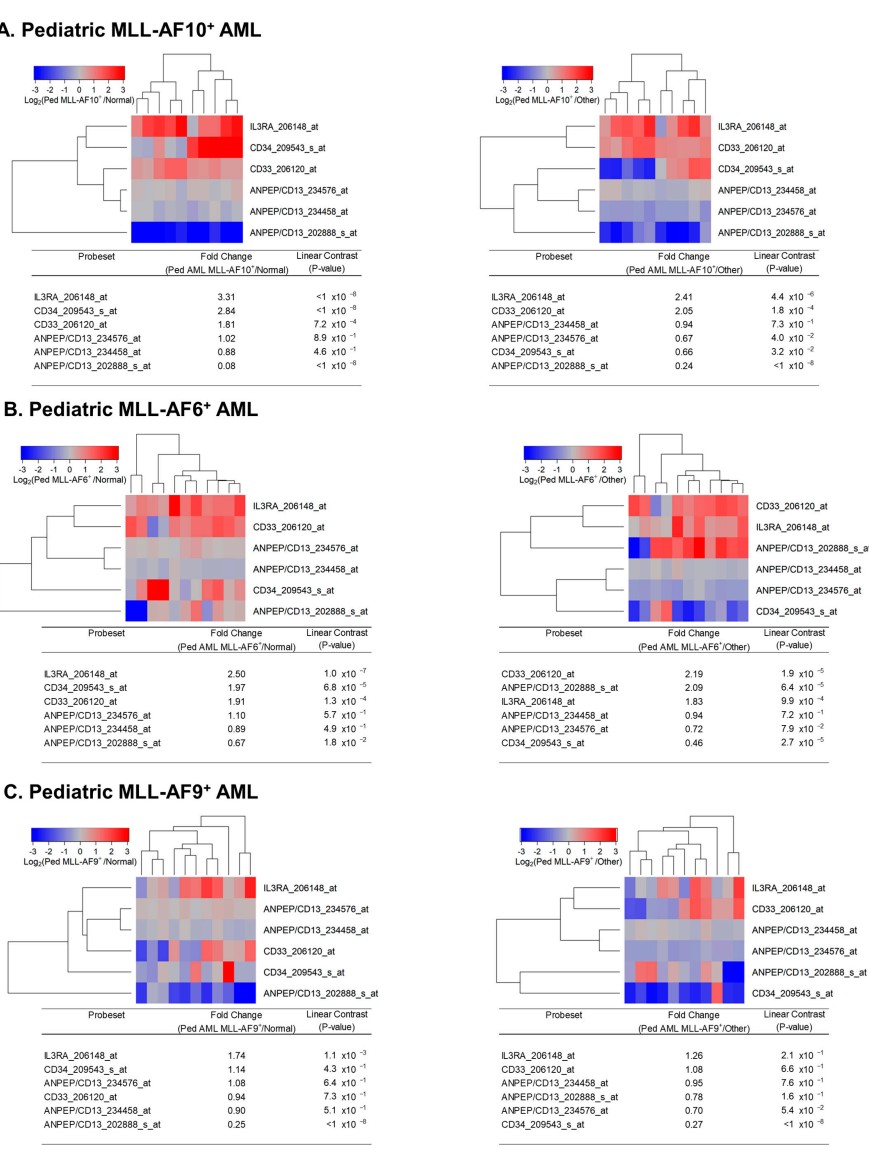

**Figure 2.** Amplified expression of IL3RA/CD123 in primary leukemic blasts from pediatric patients with MLL-AF10$^+$, MLL-AF6$^+$, and MLL-AF9$^+$ AML. We examined the gene expression data in the archived

datasets GSE13159, GSE17855, and GSE19577 and compared primary leukemic blasts from MLL-AF10$^+$ (**A**), MLL-AF6$^+$ (**B**), and MLL-AF9$^+$ (**C**) subsets of pediatric AML patients with normal hematopoietic cells from non-leukemic controls (left panels) or other AML cases (right panels). The cluster figures display the expression levels in MLL-R$^+$ AML cells mean-centered to the reference group (normal hematopoietic cells from control bone marrow samples or leukemic blasts from pediatric patients of other AML subsets) for log$_2$-transformed fold change values (blue represents underexpression and red represents overexpression in MLL-R$^+$ samples). The expression levels of coregulated probesets for both probesets (rows) and patients (columns) are organized in the depicted dendrograms. (**A**) In comparison to normal samples (N = 74; GSE13159), leukemic blasts from 10 pediatric AML patients with MLL-AF10 gene fusion (GSE19577) showed upregulated expression of IL3RA/CD123. IL3RA_206148_at was the most significantly upregulated probeset (fold change = 3.31; $p$-value < $10^{-8}$) followed by CD34_209543_s_at (fold change = 2.84; $p$-value < $10^{-8}$) (left panel). IL3RA_206148_at was significantly upregulated in MLL-AF10$^+$ pediatric AML cases (fold change = 2.41; $p$-value = $4.4 \times 10^{-6}$) when compared to other types of AML without MLL rearrangements (N = 190; GSE17855) (right panel). (**B**) Comparison of AML patients with MLL-AF6 fusion (N = 11; GSE19577) vs. normal hematopoietic cells showed that IL3RA_206148_at was the most significantly upregulated probeset (fold change = 2.50; $p$-value = $10^{-7}$) followed by CD34_209543_s_at (fold change = 1.97; $p$-value = $6.8 \times 10^{-5}$) (left panel). IL3RA_206148_at exhibited significant upregulation in pediatric AML patients with MLL-AF6 gene fusion (fold change = 1.83; $p$-value = $9.9 \times 10^{-4}$) when compared to pediatric AML without MLL/KMT2A rearrangements (N = 190; GSE17855). (**C**) IL3RA_206148_at was significantly upregulated (fold change = 1.74; $p$-value = 0.0011) (left panel) in MLL-AF9$^+$ AML cells (N = 11; GSE19577) vs. normal hematopoietic cells (left panel). However, IL3RA_206148_at exhibited an increase in expression that was not statistically significant (fold change = 1.26; $p$-value = 0.21) when compared to other types of AML without MLL rearrangements (N = 190; GSE17855) (right panel). In all 3 subgroups of AML patients with MLL/KMT2A fusions shown in (**A**–**C**), primary leukemic cells had higher CD33/CD34 ratios than primary leukemia cells from AML patients without MLL/KMT2A fusions, consistent with a relatively more advanced maturational stage and myelo-monocytic differentiation.

## A. Pediatric MLL-AF9$^+$ AML vs. MLL-AF10$^+$ AML

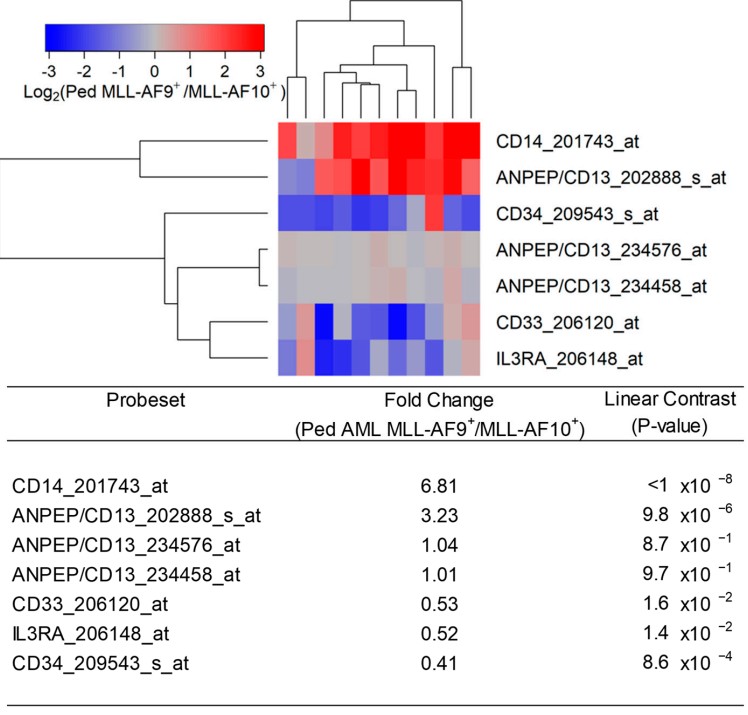

| Probeset | Fold Change (Ped AML MLL-AF9$^+$/MLL-AF10$^+$) | Linear Contrast (P-value) |
|---|---|---|
| CD14_201743_at | 6.81 | <1 x10$^{-8}$ |
| ANPEP/CD13_202888_s_at | 3.23 | 9.8 x10$^{-6}$ |
| ANPEP/CD13_234576_at | 1.04 | 8.7 x10$^{-1}$ |
| ANPEP/CD13_234458_at | 1.01 | 9.7 x10$^{-1}$ |
| CD33_206120_at | 0.53 | 1.6 x10$^{-2}$ |
| IL3RA_206148_at | 0.52 | 1.4 x10$^{-2}$ |
| CD34_209543_s_at | 0.41 | 8.6 x10$^{-4}$ |

**Figure 3.** *Cont.*

## B. Pediatric MLL-AF9⁺ AML vs. MLL-AF6⁺ AML

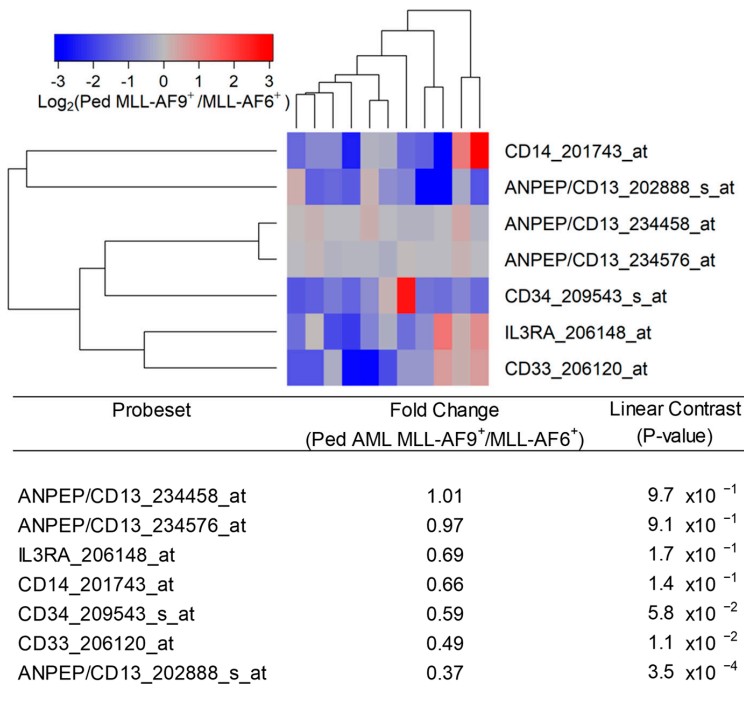

| Probeset | Fold Change (Ped AML MLL-AF9⁺/MLL-AF6⁺) | Linear Contrast (P-value) |
|---|---|---|
| ANPEP/CD13_234458_at | 1.01 | $9.7 \times 10^{-1}$ |
| ANPEP/CD13_234576_at | 0.97 | $9.1 \times 10^{-1}$ |
| IL3RA_206148_at | 0.69 | $1.7 \times 10^{-1}$ |
| CD14_201743_at | 0.66 | $1.4 \times 10^{-1}$ |
| CD34_209543_s_at | 0.59 | $5.8 \times 10^{-2}$ |
| CD33_206120_at | 0.49 | $1.1 \times 10^{-2}$ |
| ANPEP/CD13_202888_s_at | 0.37 | $3.5 \times 10^{-4}$ |

**Figure 3.** Reduced expression of IL3RA/CD123 in primary leukemic blasts from pediatric patients with MLL-AF9⁺ AML compared with MLL-AF10⁺ or MLL-AF6⁺ AML. We examined the gene expression data in the archived dataset GSE19577 and compared primary leukemic blasts from MLL-AF9⁺ AML for the differential expression of IL3RA and control genes (CD13, CD14, CD33, and CD34) with MLL-AF10⁺ AML (**A**) and MLL-AF6⁺ AML (**B**). The cluster figures display the expression levels in MLL-AF9⁺ AML cells mean-centered to the reference group (MLL-AF10⁺ or MLL-AF6⁺ AML) for $\log_2$-transformed fold change values (blue represents underexpression and red represents overexpression in MLL-AF9⁺ AML cells). The expression levels of coregulated probesets for both probesets (rows) and patients (columns) are organized in the depicted dendrograms. (**A**) In the comparison of MLL-AF9⁺ AML (N = 11) with MLL-AF10⁺ AML (N = 10), leukemic blasts from 11 pediatric MLL-AF9⁺ AML patients showed that CD14_201743_at was the most significantly upregulated probeset in these patients (fold change = 6.81; *p*-value $< 10^{-8}$). CD34_209543_s_at was the most significantly downregulated probeset in MLL-AF9⁺ AML (fold change = 0.41; *p*-value = $8.6 \times 10^{-4}$) followed by IL3RA_206148_at (fold change = 0.52; *p*-value = 0.014) and CD33_206120_at (fold change = 0.53; *p*-value = 0.016). (**B**) Comparing gene expression levels in MLL-AF9⁺ AML cells relative to MLL-AF6⁺ AML cells showed that ANPEP/CD13_202888_s_at was the most significantly downregulated probeset in MLL-AF9⁺ AML (fold change = 0.37; *p*-value = $3.5 \times 10^{-4}$) followed by CD33_206120_at (fold change = 0.49; *p*-value = 0.011). IL3RA_206148_at exhibited a 0.69-fold change (*p*-value = 0.17). This 1.45-fold reduction in expression of IL3RA in MLL-AF9⁺ AML was not statistically significant.

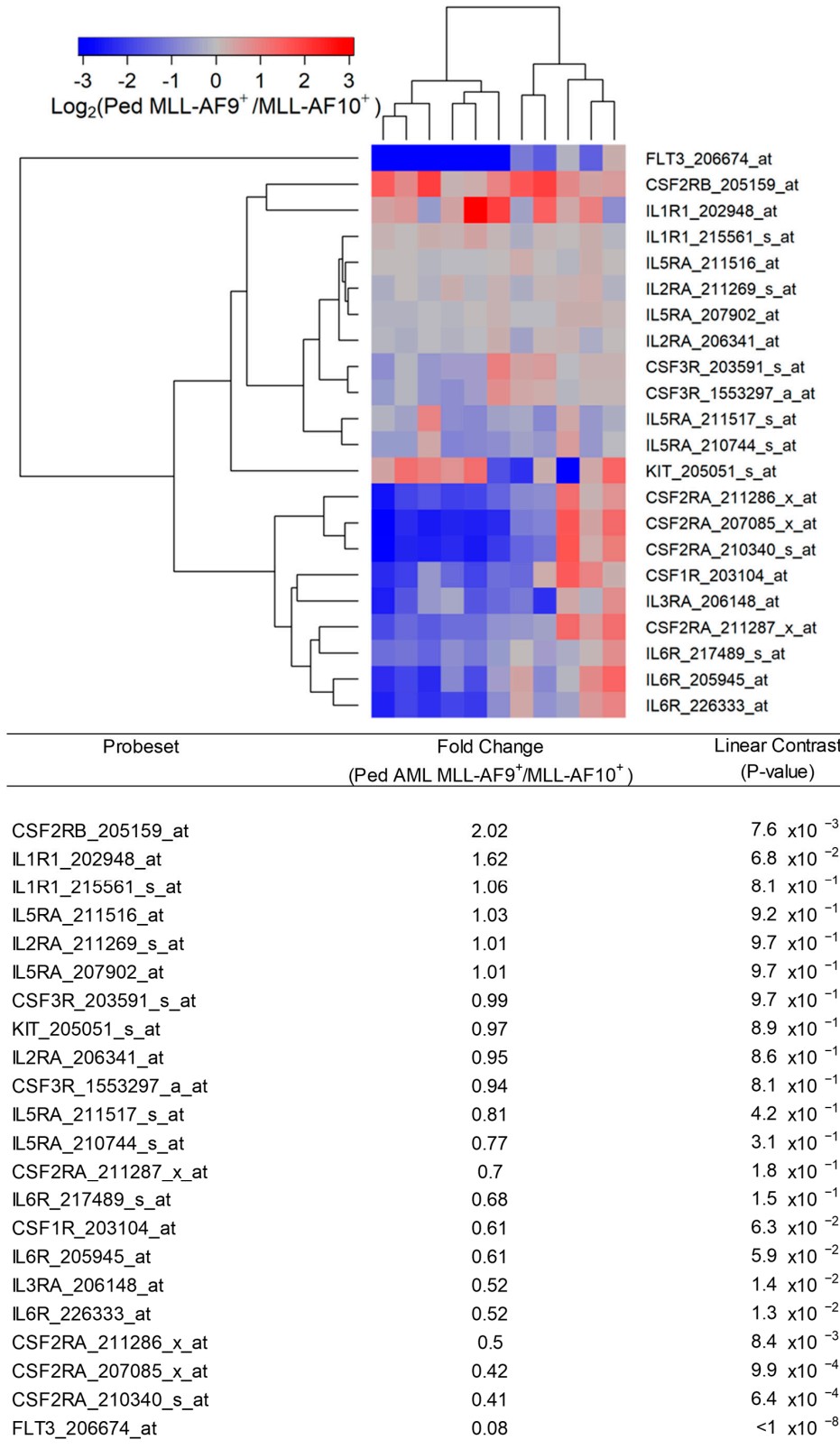

| Probeset | Fold Change | Linear Contrast |
|---|---|---|
| | (Ped AML MLL-AF9$^+$/MLL-AF10$^+$) | (P-value) |
| CSF2RB_205159_at | 2.02 | 7.6 x10$^{-3}$ |
| IL1R1_202948_at | 1.62 | 6.8 x10$^{-2}$ |
| IL1R1_215561_s_at | 1.06 | 8.1 x10$^{-1}$ |
| IL5RA_211516_at | 1.03 | 9.2 x10$^{-1}$ |
| IL2RA_211269_s_at | 1.01 | 9.7 x10$^{-1}$ |
| IL5RA_207902_at | 1.01 | 9.7 x10$^{-1}$ |
| CSF3R_203591_s_at | 0.99 | 9.7 x10$^{-1}$ |
| KIT_205051_s_at | 0.97 | 8.9 x10$^{-1}$ |
| IL2RA_206341_at | 0.95 | 8.6 x10$^{-1}$ |
| CSF3R_1553297_a_at | 0.94 | 8.1 x10$^{-1}$ |
| IL5RA_211517_s_at | 0.81 | 4.2 x10$^{-1}$ |
| IL5RA_210744_s_at | 0.77 | 3.1 x10$^{-1}$ |
| CSF2RA_211287_x_at | 0.7 | 1.8 x10$^{-1}$ |
| IL6R_217489_s_at | 0.68 | 1.5 x10$^{-1}$ |
| CSF1R_203104_at | 0.61 | 6.3 x10$^{-2}$ |
| IL6R_205945_at | 0.61 | 5.9 x10$^{-2}$ |
| IL3RA_206148_at | 0.52 | 1.4 x10$^{-2}$ |
| IL6R_226333_at | 0.52 | 1.3 x10$^{-2}$ |
| CSF2RA_211286_x_at | 0.5 | 8.4 x10$^{-3}$ |
| CSF2RA_207085_x_at | 0.42 | 9.9 x10$^{-4}$ |
| CSF2RA_210340_s_at | 0.41 | 6.4 x10$^{-4}$ |
| FLT3_206674_at | 0.08 | <1 x10$^{-8}$ |

**Figure 4.** Expression of cytokine receptors in primary leukemic blasts from pediatric patients with MLL-AF9$^+$ AML compared with MLL-AF10$^+$ AML. We examined the gene expression data in the archived dataset GSE19577 and compared primary leukemic blasts from MLL-AF9$^+$ AML patients vs. MLL-AF10$^+$ AML patients for the differential expression of IL3RA/CD123, CSF3R, KIT, CSF1R, IL2RA, IL6R, FLT3, IL1R1, CSF2RB, IL5RA, CSF2RA, and IL3RA. The cluster figure displays the expression levels in MLL-AF9$^+$

AML cells mean-centered to the reference group (MLL-AF10$^+$) for log$_2$-transformed fold change values (blue represents underexpression and red represents overexpression in MLL-AF9$^+$ AML cells). The expression levels of coregulated probesets for both probesets (rows) and patients (columns) are organized in the depicted dendrograms. In the comparison of MLL-AF9$^+$ AML (N = 11) with MLL-AF10$^+$ AML (N = 10), leukemic blasts from 11 pediatric MLL-AF9$^+$ AML patients showed significantly upregulated expression of CSF2RB_205159_at in these cases (fold change = 2.02; *p*-value = 0.0076). FLT3_206674_at was the most significantly downregulated probeset in MLL-AF9$^+$ AML patients (fold change = 0.08; *p*-value < 10$^{-8}$) followed by CSF2RA_210340_s_at (fold change = 0.41; *p*-value = 6.4 × 10$^{-4}$) and CSF2RA_207085_x_at (fold change = 0.42; *p*-value = 9.9 × 10$^{-4}$). IL3RA exhibited coregulated expression with CSF1R, CSF2RA, and IL6R (3 probesets).

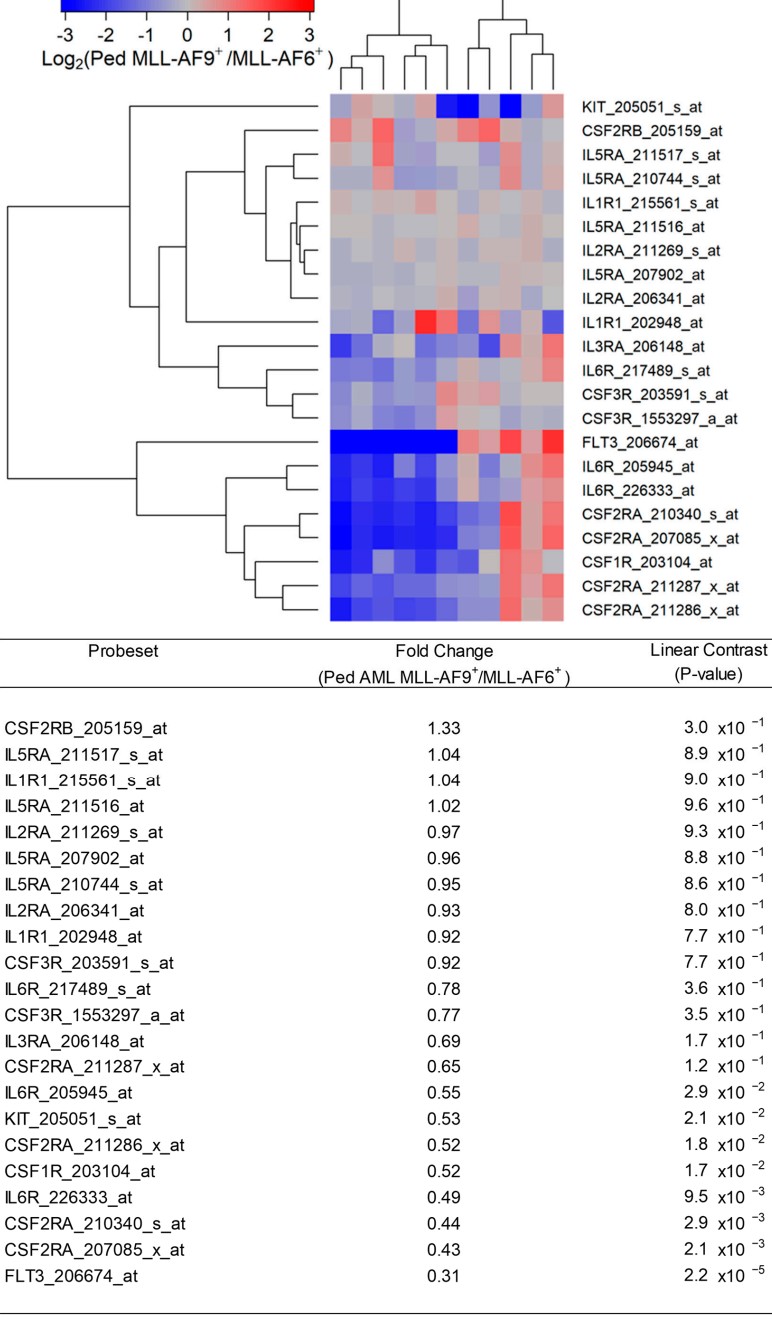

| Probeset | Fold Change (Ped AML MLL-AF9$^+$/MLL-AF6$^+$) | Linear Contrast (P-value) |
|---|---|---|
| CSF2RB_205159_at | 1.33 | 3.0 ×10$^{-1}$ |
| IL5RA_211517_s_at | 1.04 | 8.9 ×10$^{-1}$ |
| IL1R1_215561_s_at | 1.04 | 9.0 ×10$^{-1}$ |
| IL5RA_211516_at | 1.02 | 9.6 ×10$^{-1}$ |
| IL2RA_211269_s_at | 0.97 | 9.3 ×10$^{-1}$ |
| IL5RA_207902_at | 0.96 | 8.8 ×10$^{-1}$ |
| IL5RA_210744_s_at | 0.95 | 8.6 ×10$^{-1}$ |
| IL2RA_206341_at | 0.93 | 8.0 ×10$^{-1}$ |
| IL1R1_202948_at | 0.92 | 7.7 ×10$^{-1}$ |
| CSF3R_203591_s_at | 0.92 | 7.7 ×10$^{-1}$ |
| IL6R_217489_s_at | 0.78 | 3.6 ×10$^{-1}$ |
| CSF3R_1553297_a_at | 0.77 | 3.5 ×10$^{-1}$ |
| IL3RA_206148_at | 0.69 | 1.7 ×10$^{-1}$ |
| CSF2RA_211287_x_at | 0.65 | 1.2 ×10$^{-1}$ |
| IL6R_205945_at | 0.55 | 2.9 ×10$^{-2}$ |
| KIT_205051_s_at | 0.53 | 2.1 ×10$^{-2}$ |
| CSF2RA_211286_x_at | 0.52 | 1.8 ×10$^{-2}$ |
| CSF1R_203104_at | 0.52 | 1.7 ×10$^{-2}$ |
| IL6R_226333_at | 0.49 | 9.5 ×10$^{-3}$ |
| CSF2RA_210340_s_at | 0.44 | 2.9 ×10$^{-3}$ |
| CSF2RA_207085_x_at | 0.43 | 2.1 ×10$^{-3}$ |
| FLT3_206674_at | 0.31 | 2.2 ×10$^{-5}$ |

**Figure 5.** Expression of cytokine receptors in primary leukemic blasts from pediatric patients with MLL-AF9$^+$ AML compared with MLL-AF6$^+$ AML. We examined the gene expression data in the archived

dataset GSE19577 and compared primary leukemic blasts from MLL-AF9$^+$ AML patients vs. MLL-AF6$^+$ AML patients for the differential expression of IL3RA/CD123, CSF3R, KIT, CSF1R, IL2RA, IL6R, FLT3, IL1R1, CSF2RB, IL5RA, CSF2RA, and IL3RA. The cluster figure displays the expression levels in MLL-AF9$^+$ AML cells mean-centered to the reference group (MLL-AF6$^+$) for log$_2$-transformed fold change values (blue represents underexpression and red represents overexpression in MLL-AF9$^+$ AML cells). The expression levels of coregulated probesets for both probesets (rows) and patients (columns) are organized in the depicted dendrograms. Comparing gene expression changes in MLL-AF9$^+$ AML cells (N = 11) relative to MLL-AF6$^+$ AML cells (N = 11) showed that FLT3_206674_at was the most significantly downregulated probeset (fold change = 0.31; *p*-value = 2.2 × 10$^{-5}$) followed by CSF2RA_207085_x_at (fold change = 0.43; *p*-value = 0.0021) and CSF2RA_210340_s_at (fold change = 0.44; *p*-value = 0.0029). IL3RA formed a coregulated cluster of expression changes with IL6R and CSF3R (2 probesets).

### 3.2. Augmented Expression of IL3RA/CD123 in Primary Leukemic Blasts from MLL-R$^+$ Infant ALL Patients

We examined the transcript-level expression of IL3RA/CD123 in primary leukemic blasts from 80 infants with MLL-R$^+$ B-lineage ALL. IL3RA/CD123 expression in infant ALL cells was correlated with CD19 and CD34 expression levels and was 1.43-fold higher than the IL3RA/CD123 levels in normal hematopoietic cells (N = 74) (*p*-value = 3.6 × 10$^{-5}$) (Figure 6A). IL3RA/CD123 was expressed at 1.65-fold higher levels in MLL-R$^+$ infant ALL cells (N = 80) than in MLL-R$^-$ infant ALL cells (N = 17) (*p*-value = 7.1 × 10$^{-4}$) (Figure 6B). The comparison of log$_2$-transformed RMA values for leukemia cells from 48 infants with MLL-AF4$^+$ ALL (Figure 7A) as well as 16 infants with MLL-ENL$^+$ ALL (Figure 7B) with RMA values for leukemia cells from 17 infants with MLL-germline/WT (MLL-R$^-$) ALL for the probeset IL3RA_206148_at showed 1.76-fold (*p*-value = 2.1 × 10$^{-4}$) and 1.43-fold (*p* = 0.055) increases in IL3RA/CD123 expression, respectively.

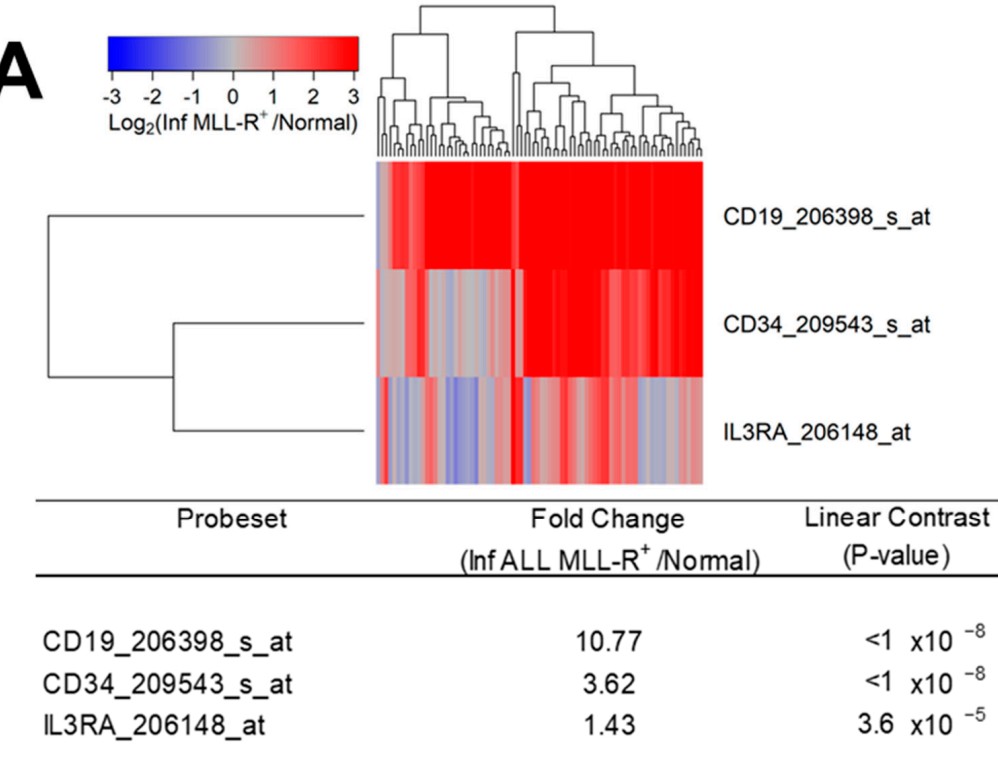

**Figure 6.** *Cont*.

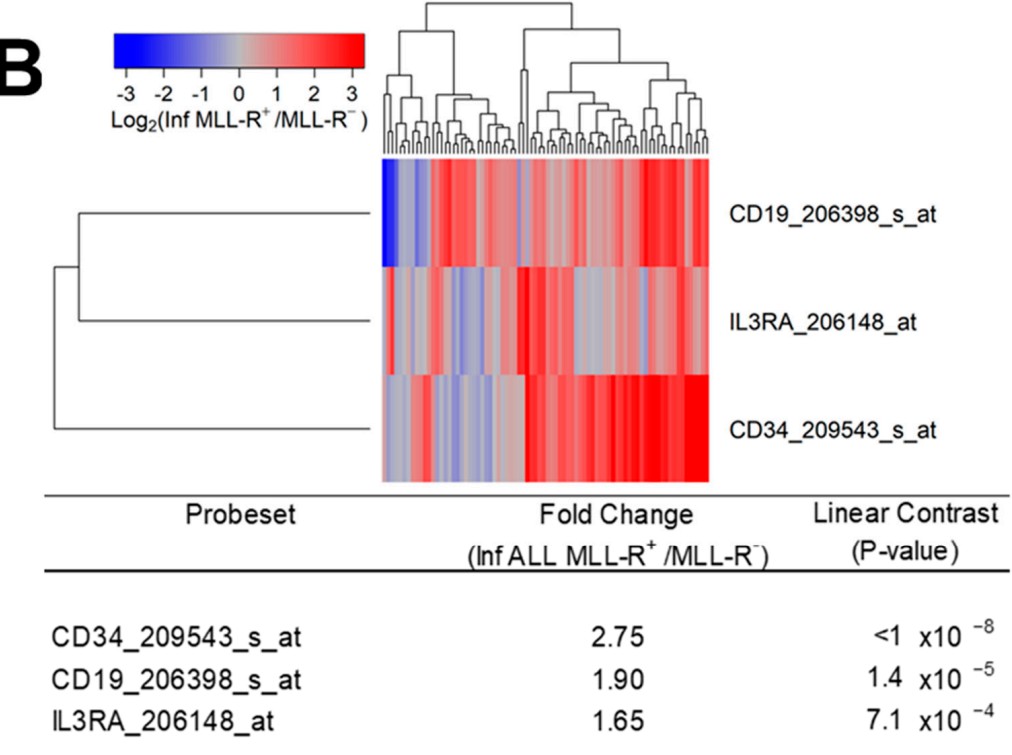

| Probeset | Fold Change (Inf ALL MLL-R$^+$/MLL-R$^-$) | Linear Contrast (P-value) |
|---|---|---|
| CD34_209543_s_at | 2.75 | <1 x10$^{-8}$ |
| CD19_206398_s_at | 1.90 | 1.4 x10$^{-5}$ |
| IL3RA_206148_at | 1.65 | 7.1 x10$^{-4}$ |

**Figure 6.** Amplified expression of IL3RA/CD123 in primary leukemic blasts from infants with MLL-R$^+$ ALL. We examined the gene expression data in the archived data infant ALL dataset GSE68720 and the control dataset from GSE13159. The cluster figures display the expression levels in MLL-R$^+$ ALL cells mean-centered to the reference group (normal hematopoietic cells from control bone marrow samples or leukemic blasts from MLL-R$^-$ infant ALL cases) for log$_2$-transformed fold change values (blue represents underexpression and red represents overexpression in MLL-R$^+$ samples). The expression levels of coregulated probesets for both probesets (rows) and patients (columns) are organized in the depicted dendrograms. (**A**) The comparison of the log$_2$-transformed RMA values for normal hematopoietic cells from 74 control samples with RMA values for leukemic blasts from 80 infants with MLL-R$^+$ ALL showed that CD19_206398_s_at was the most significantly upregulated probeset (fold change = 10.77; *p*-value < 10$^{-8}$) followed by CD34_209543_s_at (fold change = 3.62; *p*-value < 10$^{-8}$) and IL3RA_206148_at (fold change = 1.43; *p*-value = 3.6 × 10$^{-5}$). (**B**) The comparison of log$_2$-transformed RMA values leukemic cells from 80 infants with MLL-R$^+$ ALL with the RMA values for leukemia cells from 17 infants with MLL-germline/WT (MLL-R$^-$) ALL showed that CD34_209543_s_at was the most significantly upregulated probeset (fold change = 2.75; *p*-value < 10$^{-8}$) followed by CD19_206398_s_at (fold change = 1.9; *p*-value = 1.4 × 10$^{-5}$). IL3RA_206148_at exhibited a 1.65-fold increase in expression (*p*-value = 7.1 × 10$^{-4}$).

## A. Infant MLL-AF4⁺ ALL

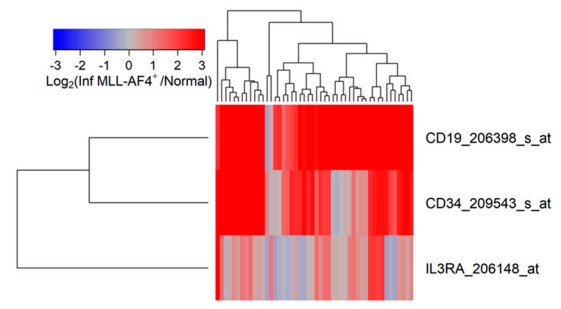

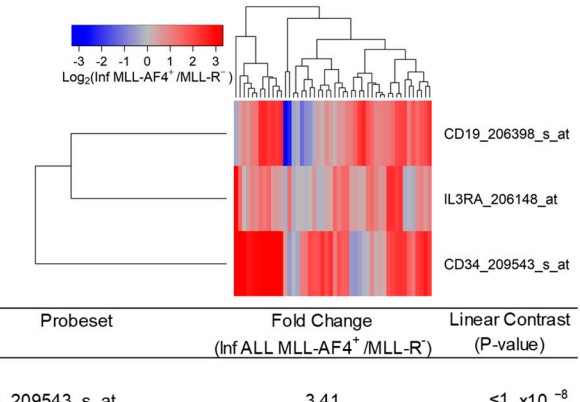

| Probeset | Fold Change (Inf ALL MLL-AF4⁺/Normal) | Linear Contrast (P-value) |
|---|---|---|
| CD19_206398_s_at | 11.10 | $<1 \times 10^{-8}$ |
| CD34_209543_s_at | 4.48 | $<1 \times 10^{-8}$ |
| IL3RA_206148_at | 1.53 | $1.5 \times 10^{-5}$ |

| Probeset | Fold Change (Inf ALL MLL-AF4⁺/MLL-R⁻) | Linear Contrast (P-value) |
|---|---|---|
| CD34_209543_s_at | 3.41 | $<1 \times 10^{-8}$ |
| CD19_206398_s_at | 1.96 | $1.0 \times 10^{-5}$ |
| IL3RA_206148_at | 1.76 | $2.1 \times 10^{-4}$ |

## B. Infant MLL-ENL⁺ ALL

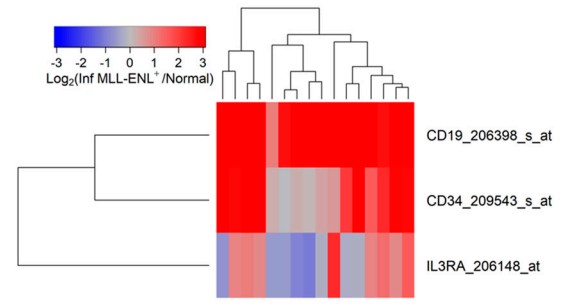

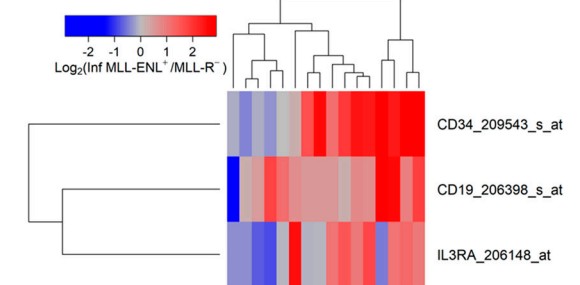

| Probeset | Fold Change (Inf ALL MLL-ENL⁺/Normal) | Linear Contrast (P-value) |
|---|---|---|
| CD19_206398_s_at | 10.64 | $<1 \times 10^{-8}$ |
| CD34_209543_s_at | 3.88 | $<1 \times 10^{-8}$ |
| IL3RA_206148_at | 1.25 | $1.3 \times 10^{-1}$ |

| Probeset | Fold Change (Inf ALL MLL-ENL⁺/MLL-R⁻) | Linear Contrast (P-value) |
|---|---|---|
| CD34_209543_s_at | 2.95 | $1.3 \times 10^{-8}$ |
| CD19_206398_s_at | 1.88 | $8.5 \times 10^{-4}$ |
| IL3RA_206148_at | 1.43 | $5.5 \times 10^{-2}$ |

**Figure 7.** Amplified expression of IL3RA/CD123 in primary leukemic blasts from infants with MLL-AF4⁺ and MLL-ENL⁺ ALL. We examined the gene expression data in the archived data infant ALL dataset GSE68720 and the control dataset from GSE13159. We compared MLL-AF4⁺ (**A**) and MLL-ENL⁺ (**B**) subsets of infant ALL patients with normal bone marrow samples (left panels) or other MLL-R⁻ ALL cases (right panels). The cluster figures display the expression levels in MLL-R⁺ ALL cells mean-centered to the reference group (normal hematopoietic cells in control bone marrow samples or leukemic blasts from MLL-R⁻ infant ALL patients) for log₂-transformed fold change values (blue represents underexpression and red represents overexpression in MLL-R⁺ samples). The expression levels of coregulated probesets for both probesets (rows) and patients (columns) are organized in the depicted dendrograms. (**A**) The comparison of the log₂-transformed RMA values for normal hematopoietic cells from 74 control samples with RMA values for leukemic blasts from 48 infants with MLL-AF4⁺ ALL showed that CD19_206398_s_at was the most significantly upregulated transcript in MLL-AF4⁺ subset of cases (fold change = 11.1; *p*-value < $10^{-8}$) followed by CD34_209543_s_at (fold change = 4.48; *p*-value < $10^{-8}$) and IL3RA_206148_at (fold change = 1.53; *p*-value = $1.5 \times 10^{-5}$) (left panel). The comparison of the log₂-transformed RMA values for leukemic cells from 48 infants with MLL-AF4⁺ ALL with the RMA values for leukemia cells from 17 infants with MLL-germline/WT ALL (MLL-R⁻) showed that CD34_209543_s_at was the most significantly upregulated probeset (fold change = 3.41; *p*-value < $10^{-8}$) followed by CD19_206398_s_at (fold change = 1.96; *p*-value < $10^{-8}$) and

IL3RA_206148_at (fold change = 1.76; $p$-value = $2.1 \times 10^{-4}$) (right panel). (**B**) The comparison of the log$_2$-transformed RMA values for normal hematopoietic cells from 74 control samples with RMA values for leukemic blasts from 16 infants with MLL-ENL$^+$ ALL showed that CD19_206398_s_at was the most significantly upregulated probeset (fold change = 10.64; $p$-value < $10^{-8}$) followed by CD34_209543_s_at (fold change = 3.88; $p$-value < $10^{-8}$). IL3RA/CD123 was not significantly augmented in the MLL-ENL$^+$ subset of infant ALL cases vs. MLL-R$^-$ infant ALL cases (fold change = 1.25, $p$-value = 0.13) (left panel). The comparison of the log$_2$-transformed RMA values for leukemic cells from 16 infants with MLL-ENL$^+$ ALL with the RMA values for leukemia cells from 17 infants with MLL-R$^-$ ALL showed that CD34_209543_s_at was the most significantly upregulated probeset (fold change = 2.95; $p$-value = $1.3 \times 10^{-8}$) followed by CD19_206398_s_at (fold change = 1.88; $p$-value = $8.5 \times 10^{-4}$). IL3RA_206148_at exhibited a 1.43-fold increase in expression that was borderline statistically significant ($p$-value = 0.055) (right panel).

## 4. Discussion

The outcome of MLL-R$^+$ AML and ALL patients, especially those with relapsed or refractory leukemia, is disappointingly poor after contemporary treatments [3,5–8,10–14,21,22]. There is an urgent need for new, more effective treatment strategies against MLL-R$^+$ AML and ALL. Menin protein, the product of the multiple endocrine neoplasia (MEN1) gene, has been shown to be critical for MLL1 function, and the Menin–MLL1 interaction plays a pivotal role in the biology of MLL-R$^+$ acute leukemias [1,41]. Small molecule inhibitors of Menin–MLL1 interaction have exhibited promising in vitro and in vivo anti-leukemic activity against leukemia cells with MLL rearrangements [42–44]. Several additional drugs are being evaluated for their activity against MLL-rearranged leukemias, including but not limited to glycogen synthase kinase 3 (GSK3) inhibitors, DNA damage response (DDR) inhibitors such as poly (ADP-ribose) polymerase (PARP) inhibitors, cyclin-dependent kinase (CDK) inhibitors, and histone deacetylase (HDAC) inhibitors [45,46]. The present study demonstrates that CD123-directed biotherapeutic agents may have clinical potential in the treatment of pediatric MLL-R$^+$ AML as well as infant MLL-R$^+$ ALL patients.

CD19 is a validated target for the treatment of B-lineage ALL patients, including those who are MLL-R$^+$ [47–52]. CD19-directed CD3-engaging bispecific antibodies and chimeric antigen receptor-carrying T-cells (CAR T) have contributed to high-quality remissions in MLL-R$^+$ ALL patients, but their success has been hampered by CD19-negative leukemic relapses [48,52] as well as lineage switching, leading to the emergence of secondary AML [48–50]. Trispecific antibodies (e.g., CD19 × CD22 × CD3) as well as bispecific CAR T (e.g., CD19 × CD22; CD19 × CD133) may be able to reduce the risk of CD19-negative relapses by concomitant targeting of more than a single leukemia-associated antigen [53–55]. Abundant expression of CD123 in MLL-R$^+$ infant ALL patients indicates that CD19 × CD123 × CD3 trispecific antibodies or sequential use of CD3 × CD19 and CD3 × CD123 bispecific antibodies could emerge as meaningful strategies against the development of CD19-negative relapses in MLL-R$^+$ ALL patients.

Previous studies have demonstrated that IL3RA/CD123 expression is amplified in AML cases with mutations in the fms-like tyrosine kinase 3 (*FLT3*) gene or the nucleophosmin (*NPM1*) gene, as well as in Philadelphia (Ph) chromosome-positive ALL cases with t(9;22)(q34;q11) translocation or Ph-like gene expression profiles [24,25,56]. Here, we extended previous studies of IL3RA/CD123 expression in acute leukemias. Our study demonstrates that CD123/IL3RA is significantly overexpressed in primary leukemic cells from MLL-R$^+$ pediatric AML as well as MLL-R$^+$ infant ALL. The upregulated expression of CD123 in pediatric MLL-R$^+$ AML and infant ALL suggests that CD123 may be a suitable target for biotherapy as part of the multi-modality treatment regimens against these difficult-to-treat subsets of acute leukemia.

The CD123 antigen is expressed not only on leukemia cells but also on normal hematopoietic progenitor cells [57–61]. Therefore, dose-limiting myelosuppression and severe neutropenia are well-established potential risks associated with CD123-directed biotherapy and limit their combination with standard myelotoxic chemotherapy. For example,

the anti-CD123 antibody drug conjugate IMGN632 [62,63] and the anti-CD123 fusion toxin Tagraxofusb [64], as well as CD123-directed CAR T-cells [65], were associated with severe neutropenia. The anti-CD123 bispecific T-cell-engaging antibody APVO436 caused severe anemia and thrombocytopenia [36]. Allogeneic TCRαβ-deficient CAR T-cells targeting CD123 caused toxicity to normal myeloid progenitor cells in preclinical settings [66]. This limitation will likely hamper the use of anti-CD123 biotherapeutics as part of frontline multi-modality regimens. That being said, this challenge could be overcome by using CD123-directed therapeutics prior to hematopoietic stem cell transplantation or applying them for post-chemotherapy consolidation with appropriate post-treatment cytokine support to prevent prolonged neutropenias.

CD123-directed biotherapeutics, including bispecific antibodies and autologous or allogeneic CAR T-cells, have the potential to cause severe cytokine release syndrome (CRS) as a dose-limiting toxicity [32,67]. This risk may be further enhanced due to the documented high-level expression of CD123 in the lungs [68]. Therefore, risk mitigation strategies will likely require specific guidelines for the management of CRS, including the use of steroids and inhibitors of the CRS-associated IL-6 pathway, such as the anti-IL6:IL6R antibody Tocilizumab [32,36,69].

MLL/KMT2A partner genes show different regulatory functions affecting the transcriptome and have been reported to drive distinct gene expression profiles and genomic alterations in pediatric AML [70–75]. MLL-AF9 fusion derived from the chromosomal translocation t(9;11) (p21–22;q23) is considered a leukemogenic event in the biology of monocytic AML, contributing to the rapid self-renewal of leukemic progenitor cells with a maturational arrest at a late stage of myeloid ontogeny via transcriptional dysregulation and aberrant activation of promoters of several target genes, including those encoding the Hox family transcription factors [71–75]. Unlike pediatric AML with MLL-AF6 and MLL-AF10 fusions that are associated with a poor prognosis, MLL-AF9[+] AML is associated with an intermediate to favorable prognosis [76–78]. In the present study, we found that primary leukemic cells from MLL-AF9[+] pediatric AML patients have significantly upregulated expression of IL3RA/CD123 when compared to normal hematopoietic cells, but they do not express IL3RA/CD123 as abundantly as primary leukemic cells from MLL-AF10[+] or MLL-AF6[+] pediatric AML patients. Unlike primary leukemia cells from MLL-AF10[+] or MLL-AF6[+] AML patients, which exhibit differentially upregulated CD123 expression compared to leukemia cells from MLL-R[−] AML patients, primary leukemia cells from MLL-AF9[+] AML patients showed a trend towards higher expression compared to primary leukemic cells from MLL-R[−] AML patients that was not statistically significant. Notably, the composite cytokine receptor profile of MLL-AF9[+] AML cells was different from that of MLL-AF10[+] or MLL-AF6[+] AML cells across multiple cytokine receptors. In addition to IL3RA/CD123, the receptors for FLT3 ligand (viz.: FLT3), GM-CSF (viz.: CSF2RA), and IL6 also showed transcript-level downregulation in MLL-AF9[+] AML cells compared to MLL-AF10[+] AML cells, whereas the gene for the common beta chain of the high-affinity receptor for IL-3, IL-5, and CSF (viz.: CSF2RB) was expressed at a higher level. Likewise, MLL-AF9[+] AML cells expressed lower levels of not only IL3RA/CD123 but also FLT3, CSF2RA/GMCSF-RA, IL6R, CSF1R, and KIT compared to MLL-AF6[+] AML cells.

At this point, we do not know if the observed differences are due to fusion-specific differences in the upregulation of the IL3RA/CD123 gene or a reflection of immunophenotypic differences corresponding to different stages of differentiation represented by the AML blasts with these particular MLL/KMT2A gene fusions (i.e., the relatively late stage of maturation of MLL-AF9[+] AML cells). If our observations in this small series are confirmed in a larger series of pediatric AML patients, mechanistic genetic experiments with clustered regularly interspaced short palindromic repeats (CRISPR) gene editing aimed at differentially knocking down the corresponding fusion proteins might provide valuable insights as to why CD123 expression in MLL-AF9[+] AML cases is not as abundant as in MLL-AF6[+] or MLL-AF10[+] AML cases [79]. The availability of de novo AML models with CRISPR/Cas9-engineered chromosomal translocations creating MLL/KMT2A fusions should facilitate

future research efforts aimed at delineating the differences in the transcriptional landscape driven by the most common MLL/KMT2A fusions in AML [80].

**Author Contributions:** Both authors have equally contributed to this manuscript. F.M.U. designed the evaluations reported in this paper, directed the data compilation and analysis, and prepared the initial draft of the manuscript. F.M.U. and S.Q. analyzed and validated data. S.Q. performed statistical and bioinformatic analyses; each author reviewed and revised the manuscript. No medical writer was involved. All authors have read and agreed to the published version of the manuscript.

**Funding:** This study received funding from Ares Pharmaceuticals. Authors F.M.U. and S.Q., who participated in the analysis and decision to submit the manuscript for publication, are affiliated with the funder.

**Institutional Review Board Statement:** Not applicable.

**Informed Consent Statement:** Not applicable.

**Data Availability Statement:** Raw Affymetrix .CEL data files on gene expression profiles of AML cells analyzed in the current study were obtained from 3 publicly available datasets deposited in the NCBI repository (GSE13159, GSE19577, and GSE17855). Raw Affymetrix .CEL data files on gene expression profiles of ALL cells analyzed in the current study were obtained from 8 publicly available datasets deposited in the NCBI repository (GSE11877, GSE13159, GSE13351, GSE18497, GSE28460, GSE7440, GSE68720, and GSE32962).

**Conflicts of Interest:** Author F.M.U. is employed by Ares Pharmaceuticals, and author S.Q. serves as a consultant for Ares Pharmaceuticals. All authors declare no other competing interests.

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
