# Peer review of "Augmented Expression of the IL3RA/CD123 Gene in MLL/KMT2A-Rearranged Pediatric AML and Infant ALL"

_onco, doi:10.3390/onco2030014_

Round 1
Reviewer 1 Report
The manuscript presents evidence from available datasets that expression of CD123 gene (IL3RA) isenhanced in MLL-rearranged pediatric aml and infant ALL. The ehanced expression of CD123 is well described in AML especialy adult AML. Here the novelty concerns the MLL-R ALL and pediatric AML that not all behave similarly with surprisingly no enhanced CD123 in the common MLL-AF9 type.
The work is well presented but descriptive. Indeed trials could have been planned already with depleting CD123+ leukemia stem cells given the previous knowledge of CD123 enhanced expression in AML. The question is how can one mechanistically present these data, why is transcription induced for CD123 in many MLL-R pediatric AML but not in MLL-AF9 and how does it unfold for ALL?
The manuscript will benefit from any mechanistic data or wet lab experiment.
Author Response
Response to Reviewer #1
Comment: The work is well presented but descriptive. Indeed trials could have been planned already with depleting CD123+ leukemia stem cells given the previous knowledge of CD123 enhanced expression in AML. The question is how can one mechanistically present these data, why is transcription induced for CD123 in many MLL-R pediatric AML but not in MLL-AF9 and how does it unfold for ALL? The manuscript will benefit from any mechanistic data or wet lab experiment.
Response: We thank the reviewer for the insightful comments and constructive criticism. In order to constructively address the peer review comments, we have added 3 new figures to better explain the findings and provide the basis for future mechanistic studies.
We also revised the Discussion section with relevant details about MLL-AF9+ AML. The revised Discussion section points out that the MLL partner genes show different regulatory functions affecting the transcriptome and have been reported to drive distinct gene expression profiles and genomic alterations in pediatric AML. MLL-AF9 fusion derived from the chromosomal translocation t(9;11) (p21–22;q23) is considered a leukemogenic event in the biology of monocytic AML contributing to rapid self-renewal of leukemic progenitor cells with a maturational arrest at a late-stage of myeloid ontogeny via transcriptional dysregulation and aberrant activation of promoters of several target genes.
The new figures Figure 3, Figure 4, and Figure 5 provide a direct comparison of MLL-AF9+ vs MLL-AF10+ and MLL-AF9+ vs MLL-AF6+ AML cells, especially relative to their relative maturational stage and cytokine receptor profiles. The Methods section has been updated accordingly. The added new text in the revised RESULTS section explains the new figures as follows:
MLL-AF9+ AML cells had 6.81-fold higher level of CD14 (P<1x10-8), 1.89-fold lower level of CD33 (P=0.016), and 2.44-fold lower level of CD34 (P=0.0086) than MLL-AF10+ AML cells, consistent with a more mature myelo-monocytic/monocytic differentiation stage. Compared to MLL-AF10+ AML cells, their IL3RA/CD123 expression was 1.92-fold lower (P=0.014) (Figure 3A). MLL-AF9+ AML cells also had lower levels of CD34 (1.69-fold; P=0.058), and IL3RA/CD123 (1.45-fold; P=0.17) than MLL-AF6+ AML cells but the differences were not statistically significant (Figure 3B).
We next asked if the IL3RA/CD123 was the only cytokine receptor that was expressed at a lower level in MLL-AF-9+ AML cells compared to MLL-AF-10+ or MLL-AF6+ AML cells and presented our results in the new figures Figure 4 and Figure 5. We found that the composite cytokine receptor profile of MLL-AF9+ AML cells was different from that of MLL-AF10+ or MLL-AF6+ AML cells across multiple cytokine receptors. The added new text in the revised RESULTS section explains the new figures as follows:
We next sought to determine if the IL3RA/CD123 was the only cytokine receptor that was expressed at a lower level in MLL-AF9+ AML cells compared to MLL-AF10+ AML cells. As shown in Figure 4, receptors for FLT3 ligand (viz.:FLT3), GM-CSF (viz.:CSF2RA), and IL6 also showed transcript-level downregulation in MLL-AF9+ AML cells compared to MLL-AF10+ AML cells, whereas CSF2RB, the gene for the common beta chain of the high affinity receptor for IL-3, IL-5 and CSF was expressed at 2.02-fold higher level (Figure 4). These results illustrate that the composite cytokine receptor profile of MLL-AF9+ AML cells differs from that of MLL-AF6+ AML cells across multiple cytokine receptors. Likewise, MLL-AF9+ AML cells expressed lower levels of not only IL3RA/CD123, but also FLT3, CSF2RA/GMCSF-RA, IL6R, CSF1R, and KIT compared to MLL-AF6+ AML cells (Figure 5).
The revised DISCUSSION section puts the results in context as follows (with references included within the manuscript):
MLL/KMT2A partner genes show different regulatory functions affecting the transcriptome and have been reported to drive distinct gene expression profiles and genomic alterations in pediatric AML (70-75). MLL-AF9 fusion derived from the chromosomal translocation t(9;11) (p21–22;q23) is considered a leukemogenic event in the biology of monocytic AML contributing to rapid self-renewal of leukemic progenitor cells with a maturational arrest at a late-stage of myeloid ontogeny via transcriptional dysregulation and aberrant activation of promoters of several target genes, including those encoding the Hox family transcription factors (71-75). Unlike pediatric AML with MLL-AF6 and MLL-AF10 fusions that are associated with a poor prognosis, MLL-AF9+ AML is associated with an intermediate to favorable prognosis (76-78). In the present study, we found that primary leukemic cells from MLL-AF9+ pediatric AML patients have significantly upregulated expression of IL3RA/CD123 when compared to normal hematopoietic cells, but they express IL3RA/CD123 not as abundantly as primary leukemic cells from MLL-AF10+ or MLL-AF6+ pediatric AML patients. Unlike primary leukemia cells from MLL-AF10+ or MLL-AF6+ AML patients exhibiting differentially upregulated CD123 expression compared to leukemia cells from MLL-fusion negative AML patients, primary leukemia cells from MLL-AF9+ AML patients showed a trend towards higher expression compared to primary leukemic cells from MLL-fusion negative AML patients that was not statistically significant. Notably, the composite cytokine receptor profile of MLL-AF9+ AML cells was different from that of MLL-AF10+ or MLL-AF6+ AML cells across multiple cytokine receptors. In addition to IL3RA/CD123, the receptors for FLT3 ligand (viz.:FLT3), GM-CSF (viz.:CSF2RA), and IL6 also showed transcript-level downregulation in MLL-AF9+ AML cells compared to MLL-AF10+ AML cells, whereas the gene for the common beta chain of the high affinity receptor for IL-3, IL-5 and CSF (viz.: CSF2RB) was expressed at a higher level. Likewise, MLL-AF9+ AML cells expressed lower levels of not only IL3RA/CD123, but also FLT3, CSF2RA/GMCSF-RA, IL6R, CSF1R, and KIT compared to MLL-AF6+ AML cells.
The revised DISCUSSION section also discusses what is currently not known and suggests future directions in research as follows:
At this point we do not know if the observed differences are due to fusion-specific differences in upregulation of IL3RA/CD123 gene or a reflection of immunophenotypic differences corresponding to different stages of differentiation represented by the AML blasts with these particular MLL/KMT2A gene fusions (i.e., the relatively later stage of maturation of MLL-AF9+ AML cells). If our observations in this small series are confirmed in a larger series of pediatric AML patients, mechanistic genetics experiments with Clustered regularly interspaced short palindromic repeats (CRISPR) gene editing aimed at differentially knocking down the corresponding fusion proteins might provide valuable insights as to why CD123 expression in MLL-AF9+ AML cases is not as abundant as in MLL-AF6+ or MLL-AF10+ AML cases (79). The availability of de novo AML models with CRISPR/Cas9-engineered chromosomal translocations creating MLL/KMT2A fusions should facilitate future research efforts aimed at delineating the differences in transcriptional landscape driven by the most common MLL/KMT2A fusions in AML (80).
The DISCUSSION section (original lines 297-308) included the following discussion about ALL which was kept unchanged in the revised manuscript:
CD19 is a validated target for treatment of B-lineage ALL patients, including those who are MLL-R+ (47-52). CD19-directed CD3-engaging bispecific antibodies and chimeric antigen receptor carrying T-cells (CAR-T) have contributed to high quality remissions in MLL-R+ ALL patients but their success has been hampered by CD19-negative leukemic relapses (47) as well as lineage switching leading to emergence of secondary AML (48-50). Tri-specific antibodies (e.g.: CD19xCD22xCD3) as well as bispecific CAR-T (e.g.: CD19xCD22; CD19xCD133) may be able to reduce the risk of CD19-negative relapses by concomitant targeting of more than a single leukemia-associated antigen (53-55). Abundant expression of CD123 in MLL-R+ infant ALL patients indicates that CD19xCD123xCD3 trispecific antibodies or sequential use of CD3xCD19 and CD3xCD123 bispecific antibodies could emerge as meaningful strategies against development of CD19-negative relapses in MLL-R+ ALL patients.
It is our considered view that the abundant expression of CD123 in MLL-R+ infant ALL patients suggests that CD19xCD123xCD3 trispecific antibodies or sequential use of CD3xCD19 and CD3xCD123 bispecific antibodies could emerge as meaningful strategies against development of CD19-negative relapses in MLL-R+ ALL patients.

Reviewer 2 Report
This is a well written and succinct manuscript that makes good use of publicly available data sets to investigate the expression levels of IL3RA/CD123 expression in pediatric AML and infant MLLr B- ALL cases. Data was clearly presented and the content was of interest to this reviewer. Minor changes are recommended but overall the manuscript is strong and would be expected to contribute to the currently available literature on this topic and general interest in CD123 as a target for emerging therapeutics.
Suggested Edits:
Would refer to KMT2A as an alternative (and in may circles more commonly used terminology for MLL gene).
Would consider adding KMT2A to the keyword list as it is not in the title or abstract currently
For figure 2 would consider adding a brief statement about CD33 expression for the subgroups – the data is presented in the figures but not commented on in the text of the manuscript or in the figure legend
Line 310 – typo - should be t(9;22)
Line 331 – typo – should read CD123
Author Response
Response to Reviewer 2
Comment #1: This is a well written and succinct manuscript that makes good use of publicly available data sets to investigate the expression levels of IL3RA/CD123 expression in pediatric AML and infant MLLr B- ALL cases. Data was clearly presented and the content was of interest to this reviewer. Minor changes are recommended but overall the manuscript is strong and would be expected to contribute to the currently available literature on this topic and general interest in CD123 as a target for emerging therapeutics.
Response: We thank the reviewer for this constructive review and recommendations. We are pleased to hear the overall positive assessment of our work.
Comment #2: Would refer to KMT2A as an alternative (and in may circles more commonly used terminology for MLL gene).
Response: We thank the reviewer for this very helpful suggestion. The recommended edit has been made. In the first sentence of the introduction, we state:
The 90-kb lysine [K]-methyltransferase 2A (KMT2A)/mixed lineage leukemia 1 (MLL) gene located at 11q23 on the long arm of chromosome 11 contains 36 exons and encodes a 431-kDa protein involved in regulation of transcription (1).
We used MLL/KMT2A instead of MLL throughout the revised INTRODUCTION.
Comment #3: Would consider adding KMT2A to the keyword list as it is not in the title or abstract currently
Response: We thank the reviewer for this very helpful suggestion. KMT2A has been added to the revised TILE, ABSTRACT, and KEYWORDS as requested.
Comment #4: For figure 2 would consider adding a brief statement about CD33 expression for the subgroups – the data is presented in the figures but not commented on in the text of the manuscript or in the figure legend
Response: We thank the reviewer for this very helpful suggestion. We have added the following detail to the revised figure legend:
In all 3 subgroups shown in [A]-[C], primary leukemic cells had higher CD33/CD34 ratios than primary leukemia cells from AML patients without MLL/KMT2A fusions, consistent with a relatively more mature maturational stage and myelo-monocytic differentiation.
We have also added a new figure, new FIGURE 3, in response to the comments of Reviewer 1, in which we compare MLL-AF9+ AML cases to MLL-AF10+ as well as MLL-AF6+ AML cases. The revised RESULTS section states:
MLL-AF9+ AML cells had 6.81-fold higher level of CD14 (P<1x10-8), 1.89-fold lower level of CD33 (P=0.016), and 2.44-fold lower level of CD34 (P=0.0086) than MLL-AF10+ AML cells, consistent with a more mature myelo-monocytic/monocytic differentiation stage. Compared to MLL-AF10+ AML cells, their IL3RA/CD123 expression was 1.92-fold lower (P=0.014) (Figure 3A).
Comment #5: Line 310 – typo - should be t(9;22); Line 331 – typo – should read CD123
Response: These typos have been corrected. We thank the reviewer for pointing out these errors.

Round 2
Reviewer 1 Report
I would accept it for publication.